# Deep Characterization and Comparison of Different Retrovirus-like Particles Preloaded with CRISPR/Cas9 RNPs

**DOI:** 10.3390/ijms241411399

**Published:** 2023-07-13

**Authors:** Max Wichmann, Cecile L. Maire, Niklas Nuppenau, Moataz Habiballa, Almut Uhde, Katharina Kolbe, Tanja Schröder, Katrin Lamszus, Boris Fehse, Dawid Głów

**Affiliations:** 1Research Department Cell and Gene Therapy, Department of Stem Cell Transplantation, University Medical Centre Hamburg-Eppendorf (UKE), 20246 Hamburg, Germany; max.wichmann@stud.uke.uni-hamburg.de (M.W.); n.nuppenau@uke.de (N.N.); a.uhde@uke.uni-hamburg.de (A.U.); ta.schroeder@uke.de (T.S.); 2Department of Neurosurgery, University Medical Centre Hamburg-Eppendorf (UKE), 20246 Hamburg, Germany; cmaire@uke.de (C.L.M.); k.kolbe@uke.de (K.K.); lamszus@uke.uni-hamburg.de (K.L.); 3Institute of Neuroanatomy, University Medical Centre Hamburg-Eppendorf (UKE), 20246 Hamburg, Germany; 4German Center for Infection Research (DZIF), Partner Site Hamburg-Lübeck-Borstel-Riems, 20246 Hamburg, Germany

**Keywords:** CRISPR/Cas, virus-like particles, iPS cells

## Abstract

The CRISPR/Cas system has a broad range of possible medical applications, but its clinical translation has been hampered, particularly by the lack of safe and efficient vector systems mediating the short-term expression of its components. Recently, different virus-like particles (VLPs) have been introduced as promising vectors for the delivery of CRISPR/Cas genome editing components. Here, we characterized and directly compared three different types of retrovirus-based (R) VLPs, two derived from the γ-retrovirus murine leukemia virus (gRVLPs and “enhanced” egRVLPs) and one from the lentivirus human immunodeficiency virus, HIV (LVLPs). First, we unified and optimized the production of the different RVLPs. To ensure maximal comparability of the produced RVLPs, we adapted several assays, including nanoparticle tracking analysis (NTA), multi-parametric imaging flow cytometry (IFC), and Cas9-ELISA, to analyze their morphology, surface composition, size, and concentration. Next, we comparatively tested the three RVLPs targeting different genes in 293T model cells. Using identical gRNAs, we found egRVLPs to mediate the most efficient editing. Functional analyses indicated better cargo (i.e., Cas9) transfer and/or release as the underlying reason for their superior performance. Finally, we compared on- and off-target activities of the three RVLPs in human-induced pluripotent stem cells (hiPSC) exploiting the clinically relevant C-C motif chemokine receptor 5 (CCR5) as the target. Again, egRVLPs facilitated the highest, almost 100% knockout rates, importantly with minimal off-target activity. In conclusion, in direct comparison, egRVLPs were the most efficient RVLPs. Moreover, we established methods for in-depth characterization of VLPs, facilitating their validation and thus more predictable and safe application.

## 1. Introduction

With the advent of designer nucleases, starting from zinc finger nucleases in the mid-1990s, followed by transcription-activator-like effector nucleases (TALEN) in 2010, targeted genome editing in living cells has become a realistic opportunity [1,2]. However, its real breakthrough came with the observation that the bacterial CRISPR-Cas system can be readily applied to cut eukaryotic DNA at specific sites guided by programmable single-guide RNAs (sgRNAs or gRNAs) [3,4]. Further developments, namely fusions of inactive Cas9 variants with various effector molecules such as deaminases, reverse transcriptase, serine integrase, and transcriptional activators or inhibitors, have formed the basis for additional editing and transient modulation techniques, i.e., base and prime editing, PASTE, CRISPRa, and CRISPRi, respectively [5,6,7,8,9,10]. Together, these new methods offer an outstanding toolbox allowing the introduction of almost any desired change in the genomes of different organisms. This has sped up basic and translational research in essentially any field of biology.

At the same time, broad application of targeted genome editing (GE) as a promising novel approach for in vivo gene therapy has been hampered by several obstacles, including immunogenicity [11] and the so far limited knowledge regarding the actual risks of unwanted side effects, e.g., due to off-target activity [12] or TP53 activation [13,14]. Moreover, as in classical gene therapy [15] efficient and targeted delivery of GE tools to the tissue of choice remains the most important limitation for broad application of genome editing in vivo in human beings. To mediate the desired genetic change, GE systems (e.g., Cas9, and sgRNA(s)) have to be delivered to target cells either in the form of nucleic acids encoding the information for their expression in the target tissue or directly as active proteins (for CRISPR-Cas—as ribonucleoprotein complex (RNP)). Given their modes of action, correction of monogenic diseases is the most obvious application of in vivo gene therapy with the different CRISPR-based GE tools, with the liver and eye being the “easiest” target organs [16].

It is critical to acknowledge that the classical attributes of “ideal” vector systems for gene therapy [15] do not necessarily apply to the delivery of CRISPR-Cas components. While some requirements, such as efficient packaging, cargo protection, tissue-specific delivery, and expression, remain unchanged [16], others are fundamentally different. Most importantly, in order to decrease the risk of off-target activity, expression of GE systems in the target cells should be limited with regard to both magnitude and duration [17]. This implies that standard virus-based vectors successfully used in human gene therapy cannot readily be used for the delivery of GE tools in patients [18]. Indeed, broadly used γ-retro- and lentivirus-based vectors integrate into the host genome and thus confer long-term expression, a sine qua non for many classical gene-therapy strategies. In contrast, the most popular DNA-virus vectors (adenoviral [AV] and adenovirus-associated [AAV] vectors) are normally present episomally in target cells, although there is a relatively high risk in the range of 0.1% of accidental integration [19,20]. On the downside, the latter vectors are regularly applied at huge multiplicities of infection, normally resulting in high transgene expression levels, which remain relatively stable in non-dividing cells. Moreover, the double-strand breaks induced by the nuclease could be expected to increase the risk of unwanted vector integration. Altogether, novel tools allowing delivery of GE tools as short-lived molecules, such as mRNA or, preferentially, proteins/RNPs are urgently needed.

Several delivery methods have been proposed and optimized to facilitate efficient and safe in vivo GE. To address the above limitation of viral vectors, a variety of non-viral vectors have been recently developed [21], with particular focus on lipid nanoparticles (LNPs) [22,23,24,25]. However, virus-derived vectors remain an attractive tool, since viruses have to evolve highly efficient strategies for cell entry and delivery of their genetic information into different target tissues. In order to adopt retroviral vectors for GE, they were engineered to deliver only mRNAs not capable of integration [26,27]. More recently, a new class of vectors, referred to as virus-like particles (VLPs), was introduced for the delivery of RNPs [22,28,29]. By delivering an RNP cargo, VLPs are designed to combine the expectedly better safety profile of ultra-short expression of the CRISPR/Cas components in the target cells with the high transduction efficiency of virus-derived vectors [23].

To ensure that VLPs contain GE tools, they are produced in dedicated producer cells overexpressing, e.g., the CRISPR-Cas system components. In addition, various strategies have been employed to facilitate the efficient loading of the editing machinery, namely CRISPR-Cas RNPs, into the VLPs. For example, several groups have fused the Cas nuclease to the retroviral gag protein to generate retrovirus-like particles (RVLPs) [30,31,32] (Figure 1A). Production and purification steps for diverse RVLPs appear to be similar, but so far, no direct comparison of the different particles and their GE efficiencies has been reported. Here, we aimed to characterize and compare three different types of Cas-RNP preloaded RVLPs, two (gRVLPs and “enhanced” egRVLPs) derived from the γ-retrovirus *murine leukemia virus* (MLV) and one (LVLPs) from the lentivirus *human immunodeficiency virus* (HIV). To do so, we first unified and optimized their production processes. Addressing different target loci, we observed substantial differences in their editing efficiencies, which led us to thoroughly characterize the three RVLPs. For this reason, we first adopted techniques commonly used for the characterization of extracellular vesicles (EVs) to study RVLP morphology. Next, we correlated their GE potential with RVLP concentration and the amounts of total as well as functionally active Cas9 in the particles. Finally, we compared on- and off-target activities of the three RVLPs in human-induced pluripotent stem cells (hiPSC) exploiting the clinically relevant C-C motif chemokine receptor 5 (CCR5) as the target. Together, our data suggest that in-depth characterization of RVLPs is indispensable for their validation and comparison to ensure predictable and safe application.

## 2. Results

### 2.1. Establishing Analogous Production Conditions for Three Different Retrovirus-like Particles

The recently designed retrovirus-derived particles—γ-retrovirus-like particles (gRVLP), lentivirus-like particles (LVLP), and enhanced γ-retrovirus-like particles (egRVLP)—have all been proven to efficiently pack and deliver CRISPR/Cas complexes to target cells, thus facilitating genome editing [30,31,32]. Even though all three systems are based on retroviruses, they have been designed in distinctive ways. gRVLP uses the classical alignment of the *Streptococcus pyogenes* Cas9 (SpCas9) gene with two nuclear localization signals (NLSs) surrounding the nuclease coding sequence [30]. For LVLP design, Hamilton et al. removed the NLS from the N-terminus and placed another one on the C-terminus of SpCas9 [31]. The most advanced and customized design can be found in the construction of egRVLP; in this case, the authors optimized the linker sequence recognized by Moloney MLV (M-MLV) protease and added three nuclear export signals (NESs) to the linker’s C-terminus [32]. Such a configuration was claimed to enhance both cytoplasmic localization of the SpCas9 necessary for efficient Cas9 packing to the VLPs in the producer cells and nuclear localization of the nuclease after its delivery to the target cell (following the cleavage of NESs) [32]. A schematic overview of the three different designs, including the used linker sequences, is provided in Figure 1A.

For initial characterization and comparison of gRVLPs, LVLPs, and egRVLPs, we started by unifying and optimizing their production methods (the original procedures involved different transfection strategies). We used our standard calcium-phosphate-based protocol (for details, see the Section 4). First, we assessed the potential influence of increasing the GagCas9-to-GagPol ratio (Table 1) on the morphology of the different RVLPs (Figure 1B) and on the respective GE efficiencies (Figure 1C).

To do so, for each individual RVLP, we used increasing GagCas9-to-GagPol plasmid ratios (each five for the gRVLP and LVLP and two for egRVLP, Table 1) including the ones initially reported [30,31,32]. All RVLPs were pseudotyped with the G protein of vesicular stomatitis virus (VSV-G) and produced by cells expressing Cas9 and GFP-directed sgRNAs. To allow meaningful comparison, each RVLP production was repeated three to six times in independent settings.

### 2.2. Analogously Produced Cas9-Preloaded Retrovirus-like Particles (RVLPs) Mediate Different Genome-Editing Efficiencies

In the next step, we aimed to characterize the RVLP produced using the various plasmid ratios indicated in Table 1. To do so, we exploited nanoparticle tracking analysis (NTA) to determine the sizes and numbers of particles. In addition, we applied identical as well as increasing volumes of individual vector preparations to assess the capacity of the different RVLPs to mediate GE in a GFP-transgenic 293T-derived reporter cell line. The GE rates obtained were set in relation to the numbers of measured VLPs (Appendix A).

For gRVLPs, we used one GagCas9-to-GagPol plasmid ratio (1:3) that was lower than in the original publication [30], the original (2:3), and three increased ratios (10:9, 20:9, 10:3) (Table 1). The supernatants obtained (n = 4 for each condition) were analyzed by NTA. There was some apparent tendency towards smaller RVLP sizes at elevated plasmid ratios, but the variance between individual readings was quite high (Figure 1B). For each of the five groups, several independent transduction rounds were performed using the same volumes of gRVLP-containing supernatants. GE rates in reporter cells were determined by flow cytometry. As shown in Appendix A, GE rates were directly correlated with gRVLP numbers for all five groups. However, maximal GE rates were significantly higher at increased GagCas9-to-GagPol ratios (groups 4–5 vs. groups 1–3, *p* < 0.01, Table 1; Figure 1C), with a maximum at the 10:3 ratio (group 5; Figure 1C). Of note, whereas the numbers of gRVLP as measured by NTA directly correlated with GE efficiencies within each of the five groups, they were not directly comparable between the groups (Appendix A).

To optimize LVLP production, we used a slightly modified strategy: Together with the four GagCas9-to-GagPol plasmid ratios (1:4, 1:1, 2:1, 5:1) suggested in the original work [32], we tested their most efficient ratio (2:1) with 2.5 times higher absolute amounts of both plasmids. The latter group was based on the observation that gRVLP produced with a 3 µg Gag-Cas9 plasmid mediated the highest editing rates. Due to the observed very low GE rates obtained with initially used vector productions in 6-well plates, the protocol was scaled up to 100-mm dishes (for groups 2–5 only); plasmid amounts were adjusted to the growth area. Using concentrated supernatants, we observed definitive GE rates in 293 T cells (Figure 1C). Increased GagCas9-to-GagPol plasmid ratios (2:1 and 5:1) did not result in changes in particle sizes (Figure 1B), but again in higher GE efficiencies. As evident from Figure 1C, however, the maximal GE rate obtained with LVLPs (8.9%) was almost 10 times lower than that mediated by optimal amounts of gRVLPs (88.1%), even though the same target sequences were used. As in the case of gRVLP, our analysis showed that GE rates clearly correlated with the number of applied LVLPs within each group, whereas no direct comparison was possible between the five groups (Appendix A).

Considering the above results, for production of egRVLP, we focused on two different GagCas9-to-GagPol ratios—the amounts used in the original publication (2:5 ratio) and the 10:3 ratio with 3 µg GagCas9 plasmid found to be optimal for gRVLPs. Again, there seemed to be some tendency towards smaller sizes of purified particles for the higher ratio (group 2, Figure 1B). Using the increased amount of GagCas9 plasmid and the ratio following the gRVLP experience, we produced egRVLP that mediated up to 87.7% GE efficiency, compared to up to 72.7% GE obtained with egRVLP produced with the original amounts of plasmids (Figure 1C). Importantly, for both groups of egRVLP, we observed a very high batch-to-batch consistency (Appendix A) of produced VLPs. This is exemplarily illustrated by a comparison of the mean values of GE rates obtained with identical plasmid amounts (group 5 for gRVLP and group 2 for egRVLP; Table 1) calculated based on data in Appendix A. In fact, the mean GE rates mediated by tested egRVLP batches were 68.5% compared to 34.7% for gRVLP batches.

### 2.3. Efficient Genome Editing of C-C-Motive-Chemokine-Receptor 5 (CCR5) Using RVLPs

In the next step, we wanted to test whether the same pronounced differences in GE efficiencies mediated by the three RVLP types would apply to another target. To this end, we selected the C-C motif chemokine receptor 5 (CCR5) gene. CCR5 is a clinically relevant target since it serves as a co-receptor of CCR5-tropic human immunodeficiency virus 1 (HIV1), which mediates the primary HIV infection of CD4^+^ cells in the vast majority of patients. Previously, CCR5 knockout was successfully achieved using zinc finger nucleases, TAL effector nucleases, and CRISPR/Cas [33,34,35,36]. Using a dedicated TALEN, our group showed high-efficiency CCR5-knockout and protection of gene-edited CD4 T cells from infection with CCR5-tropic lentiviral vectors and replication-competent CCR5-tropic HIV-1-BaL strains [26,37,38].

To target CCR5 with CRISPR/Cas9, we initially designed four different CCR5-targeting gRNAs (Appendix A). All gRNAs were cloned into an all-in-one LeGO CRISPR/Cas [C/C] vector [39] to be expressed together with Cas9. The resulting four lentiviral LeGO-CC vectors were used to transduce 293T cells at low MOI to ensure the presence of single copies of the CRISPR/Cas cassette inserted into their genome [40]. Transduction rates were determined by flow cytometry based on the co-expressed GFP, and *CCR5* gene editing was assessed with gene-editing frequency digital PCR (GEF-dPCR) [37] (design presented in Appendix A). The range of editing rates at the four target loci varied between 11–80% of the transduced cells (Figure 2A). Based on the obtained data, the gRNA assigned as C-1 was further used to program gRVLP, LVLP, and egRVLP particles.

VSV-G pseudotyped Cas9/C1 VLPs of all three types were produced as described above. Quantification with NTA indicated high particle concentrations for all purified VLP batches. For maximal comparability, we initially applied very similar numbers of particles for all three VLP types. However, since egRVLP mediated GE in essentially 100% of the alleles in the first experiments, for this vector, we repeated transductions with half the dose. The actual particle numbers used are indicated in Figure 2B. All different C-1 programmed vectors, gRVLP, LVLP, and egRVLP, were able to knock out CCR5 in 293 T cells, albeit with different efficiencies (Figure 2C). Strikingly, even at the reduced dose, egRVLP still knocked out almost all CCR5 alleles (Figure 2C). Notably, as in the case of GFP-targeting application of the C-1 programmed LVLP resulted in the lowest GE rate (Figure 2C).

### 2.4. No Major Impact of RVLP Type on Particle Morphology and Tetraspanin Composition

Next, we asked whether there are differences in the morphology of VLPs that potentially underlie their diverse transduction capacity. To address this, we again produced and purified VSV-G pseudotyped, GFP-programmed gRVLPs, LVLPs, and egRVLPs. First, we used aliquots of different RVLP batches for NTA analysis. The obtained data (Appendix A) confirmed that all VLPs had the anticipated particle sizes between 110 nm and 130 nm with no significant differences between groups (compare Figure 1B). Notably, medium collected from producer cells that underwent mock transfection or did not undergo transfection at all contained in the mean 3 times less particles of the expected size (Appendix A). This is consistent with the expectation that cells constantly produce extracellular vesicles that have similar sizes as RVLPs.

In order to obtain more detailed characteristics of the different particles, we employed multi-parametric imaging flow cytometry (IFC) to measure CD9, CD63, and CD81 tetraspanin markers on the surface of gRVLP vs. LVLP. These markers are considered to be specific EV markers that are ubiquitously present on EVs in most cell types [41,42]. Previously, IFC analysis was successfully used to distinguish different subpopulations of EVs based on tetraspanin surface profiles [43]. Our analysis showed that CD9, CD63, and CD81 were ubiquitously present on the surface of the majority of particles for both gRVLP and LVLP samples (Figure 3A and Appendix A). Consequently, with regard to their tetraspanin surface profiles, our purified VLP preparations looked quite homogenous without apparent subpopulations (Figure 3B). Consequently, with regard to their tetraspanin surface profiles, our purified VLP preparations looked quite homogenous without apparent subpopulations (Figure 3B).

Together, data from NTA and IFC analysis is indicative of a high purity of RVLPs with relatively low “contamination” by classical EVs. In addition, we did not observe major differences between the three studied RVLP systems in terms of morphology and surface markers.

### 2.5. RVLPs’ Ability to Mediate Genome Editing Correlates with Their Cas9 Content, but Varies between the Tested Retrovirus-like Particles

Given these results, we aimed to verify whether the observed differences in the RVLPs’ potential to facilitate editing might be due to discrepancies in the RNP packaging process. First, we used IFC to check if we could detect Cas9 in the VLPs. To this end, we labeled the VLPs with antibodies against Cas9 and a mixture of anti-tetraspanin (CD9, CD63, and CD81) antibodies. As evident from Figure 4A (left dot plot), the Cas9 signal indeed almost completely co-localized with the tetraspanin signals, suggesting the presence of Cas9 inside the VLPs. To confirm that the Cas9 was bound to the RVLP capsid, we added a lysis step during IFC sample preparation. As previously shown for EVs [43], the lysis step led to membrane destruction, resulting in complete loss of the tetraspanin signals (Figure 4A, right dot plot). In striking contrast, the Cas9 signal remained almost unchanged after lysis in comparison with single-marked RVLPs (Figure 4A, right vs. middle dot plot).

In order to quantify the Cas9 content in the different VLPs, we optimized a previously published ELISA assay [44]. A detailed description of the improved protocol is given in the Section 4. Using this assay, we obtained data that enabled us to calculate the number of Cas9 molecules per VLP. Unexpectedly, we did not observe significant differences between the three tested systems. If at all, there was a trend towards lower Cas9 content in the egRVLPs (Figure 4B), which we found to mediate the highest editing rates. These data exclude reduced CRISPR/Cas9 RNP packaging in LVLPs as the underlying reason for their profoundly lower editing efficiencies.

To further investigate the impact of Cas9 concentration on actual editing activity, we performed serial transduction experiments and related Cas9 protein amounts to the measured GE rates (Figure 4C). We found a direct and highly significant correlation (gRVLP: R^2^ = 0.794, *p* < 0.0001; LVLP: R^2^ = 0.857, *p* < 0.0001; egRVLP: R^2^ = 0.830, *p* < 0.0001) between GE efficiencies and Cas9 amounts for each of the three VLP types. At the same time and in line with the above data, we again observed substantial differences in the actual capacity of gRVLPs, LVLPs, and egRVLPs in mediating genome editing (Figure 4C). In fact, with both types of gamma-retrovirus-based VLPs, approximately 10 times higher maximal editing rates were obtained at much lower Cas9 concentrations as compared to LVLPs. At the same time, egRVLP performed much better than gRVLPs. Interestingly, in some experiments with egRVLP, we obtained high-rate GE with Cas9 amounts below the ELISA detection level (Figure 4C).

Together, these data point to much greater functional activity of Cas9 protein transferred using gRVLPs and, particularly, egRVLPs. Previously, Banskota et al. [32] showed that the configuration used in egRVLPs with additional nucleus export signals (NESs) resulted in significantly increased cytoplasmic localization of gag-Cas fusion proteins, most probably facilitating more efficient packaging. In addition, the used configuration and the different sequences of the protease cleavage sites could enable better cargo release after transduction. In any case, we proved the substantial impact of the type and design of RVLPs on their performance as GE vectors. Moreover, used in concert, the two parameters—Cas9 content and VLP numbers—might be very handy to predict GE rates mediated by retrovirus-based VLPs.

### 2.6. Efficient Gene Editing in Human Induced Pluripotent Stem Cells (hiPSC) Using VLPs

In the concluding set of experiments, we aimed to compare the usefulness of the three types of VLPs programmed to knock out a clinically relevant target gene (*CCR5*) in a cell type with high potential for therapeutic application, namely hiPSCs. To address this point, we compared the on-target efficiency of GE mediated by gRVLPs, LVLPs, and egRVLPs, as well as their potential off-target activities at difficult-to-predict sites.

For CCR5 knockout, we made use of the previously selected CCR5-specific C-1 gRNA (Appendix A). In addition, we designed a novel gRNA harboring a single-nucleotide deletion in the sequence of C-1 gRNA (C-1del, Appendix A). This approach was based on earlier findings that gRNAs might also bind to off-target sequences containing single-nucleotide indels compared to their actual target sequence [13,45,46]. Importantly, such indel-bearing off-targets are not predicted by the majority of commonly used software tools [13].

C-1 and C-1del programmed VSV-G pseudotyped RVLPs of all types were produced and purified as above. We first performed transductions using similar numbers of C-1 programmed gRVLP, LVLP, and egRVLP transferring comparable amounts of Cas9, as measured by NTA and ELISA, respectively (Figure 5A). CCR5ko rates were assessed with the same GEF-dPCR reaction as before. As previously (compare Figure 2), C-1 programmed egRVLP mediated the by far highest editing rates (up to 100%) (Figure 5A, right panel), even though according to the ELISA data, they contained the lowest amount of Cas9 protein among the three different CCR5-specific RVLPs (Figure 5A, middle panel). We also observed efficient CCR5 knockout for both C-1 programmed gRVLPs and LVLPs. Notably, in the hiPSC, gRVLPs (mean 34.7%) and LVLPs (41.0%) mediated almost identical editing rates (Figure 5A, right panel), which was in some contrast to the data obtained with 293T cells (Figure 2C) but might be due to the approximately 2 times higher number of LVLPs compared to gRVLPs (Figure 5A, left panel).

We then repeated the hiPSC transduction experiment with the C-1del programmed RVLPs. Applied particle numbers and Cas9 amounts were in the same range as for the C-1 programmed RVLPs (Figure 5B, left and middle panels). We used high-sensitivity GEF-dPCR to quantify off-target cutting in the *CCR5* locus with thresholds set based on the hiPSC mock control (zero signals). For all three RVLPs, we observed very low levels (0–3 positive droplets in individual samples) of off-target activity with an estimated range of 0.07 to 0.17% with no significant differences (Figure 5B, right panel).

Finally, for convenient comparison, we calculated the number of VLPs and Cas9 molecules applied per edited cell for both 293T and iPS cells (experiments resulting in 0% GE were excluded from this analysis). As shown in Appendix A for the CCR5 target locus in two different cell types, the data obtained, though limited, confirmed that egRVLPs were most efficient in mediating editing.

In conclusion, our data obtained in 293T as well as primary human iPS cells confirms the high potential of retrovirus-based virus-like particles for targeted genome editing. Moreover, the results of our direct comparison indicate that egRVLPs are significantly more efficient in mediating on-target editing than gRVLPs and LVLPs. Notably, this higher efficiency is mediated by lower amounts of transferred Cas9 protein. In a model experiment, we found equally low off-target activities for all three VLP types. However, these data need to be confirmed using more detailed analyses and exploring different target sites.

## 3. Discussion

With the advent of CRISPR/Cas targeted genome editing has become not only an efficient instrument in basic research, but a realistic option for high-specificity gene therapy. More recently, novel genome editing tools based on deactivated Cas proteins coupled with other effector molecules have opened even further possibilities, such as CRISPRa and CRISPRi, base, and prime editing, and PASTE [5,6,7,8,9,10]. VLPs and EVs have already been successfully used to deliver different types of GE systems [47,48]. Based on their features, retrovirus-based VLPs have been suggested as promising delivery tools for GE components, particularly in the context of therapy for monogenic diseases [30,31,32,49,50]. An example of the usefulness of egRVLPs for therapeutic in vivo base editing was recently provided in a mouse model of Leber congenital amaurosis by Banskota et al. [32]. Highly specific, targeted delivery of GE tools as achievable with virus-derived vectors would potentially also open the possibility to use them for cancer gene therapy [51]. At the same time, RVLPs mediate the transient transfer of the nucleic acids (e.g., sgRNA) and proteins (e.g., Cas9) necessary for editing and thus avoid or at least minimize inherent problems associated with high-level and/or long-term expression of GE systems [52]. These problems include increased off-target activity [53,54], p53 activation [14] and cell cycle arrest [55], as well as elicitation of an immune response against transferred proteins [11].

Different types of retrovirus-based VLPs, including γ-retrovirus-like particles (gRVLPs), lentivirus-like particles (LVLPs), enhanced γ-retrovirus-like particles (egRVLPs), and chimeric γ-retrovirus-like particles have been introduced [30,31,32,55]. High-efficiency editing rates were reported for all types of VLPs, but so far, their performance has not been directly compared. In this work, we set out to comparatively analyze the editing efficiencies and off-target activities of three different retrovirus-based virus-like particles—gRVLPs, LVLPs, and egRVLPs. All these RVLPs are based on the same principle—direct fusion of Cas9 to the retroviral gag protein. To ensure maximal comparability, we unified the VLP production for all three RVLP types. We also optimized the GagCas9-to-Gagpol ratio in the producer cell line by adjusting the plasmid concentrations for all three VLPs. Based thereon, we were able to produce high RVLP numbers, as was confirmed using dedicated assays. The latter included nanoparticle tracking analysis (NTA) and imaging flow cytometry (IFC), which we adapted in order to visualize, characterize, and quantify the produced RVLPs. This allowed us to show that the particles produced were of the expected sizes, intact, and showed similar characteristics, including surface composition. Moreover, we confirmed that similar numbers of RVLPs were used in the transduction experiments.

To compare their potential for mediating genome editing, the different RVLPs were programmed with identical gRNA targeting sequences in marker as well as clinically pertinent genes and tested on various target cell systems, including highly relevant hiPSC. Independent of the target sequences, we found substantial differences in the RVLPs’ ability to facilitate targeted gene knockout as measured by flow cytometry and/or dedicated high-sensitivity GEF-dPCR assays [37,56]. Most importantly, egRVLPs were consistently found to confer the highest knockout rates, with essentially 100% knockout of the therapeutically relevant CCR5 gene in the 293T cell line, but also in hiPSCs that are of high clinical relevance. Altogether, gRVLPs mediated better on-target performance than LVLPs, even though there was almost no difference in targeting CCR5 in hiPSCs. For all three RVLP types, we found low-level off-target activity (at the detection limit of GEF-dPCR) using a gRNA lacking one single nucleotide compared to the actual target sequence. This approach to discovering off-target activity is not only very sensitive, but also highly useful, since indel-containing off-targets are not predicted by the majority of the commonly used software tools [13]. Importantly, the very low-level off-target activity found for all three RVLPs could most probably be eliminated by replacing conventional spCas9 with next-generation high-fidelity variants, such as eSpCas9 1.0 [13]. In summary, our comparative data indicate that egRVLPs represent the most efficient and promising of the three different VLP systems.

We next set out to uncover the potential reasons underlying the observed substantial differences between the three systems. As noted above, the IFC did not reveal major disparities in the membrane composition of the VLPs. Moreover, using an improved ELISA method, we were able to show that the content of Cas9 per RVLP did not differ significantly (*p* = 0.139) between the three RVLP types. In fact, there was even a tendency toward a lower Cas9 content in the egRVLPs. This data suggests that Cas9 delivered by egRVLP is functionally more active and or more accessible. To improve the packaging of Cas9 into VLPs in the producer cell line, Banskota et al. [32] had added 3 nuclear export signals to the Cas9 (Figure 1). However, using our improved ELISA protocol, we did not observe substantial differences in the absolute Cas9 load between the three VLP types. A further difference between the γ-retrovirus- and lentivirus-based VLPs is the localization of the two nuclear localization signals (NLSs) (Figure 1). In the gRVLPs and the egRVLPs, the Cas9 proteins are fused at their N- and C-termini to bipartite NLSs, whereas in the LVLPs, both NLSs are located at the C-terminus. It could be possible that the latter configuration results in impaired nucleus entry compared to the one used in the γ-retroviral VLPs. Altogether, we were able to corroborate the finding of Banskota et al. [32] that their optimized vector configuration mediated higher knockout rates, indicating improved transfer, release and/or functionality of the delivered Cas9. Moreover, we could show that both the number of particles and Cas9 molecules per edited (knockout) allele/cell were substantially lower for egRVLPs compared to gRVLPs and LVLPs.

## 4. Materials and Methods

### 4.1. Cell Culture

HEK293T cells (CRL-3216 ATCC; Manassas, VA, USA), and their derivatives were cultured in Dulbecco’s modified Eagle’s medium (DMEM; Glutamax, Gibco/Life Technologies, Carlsbad, CA, USA) supplemented with 10% fetal calf serum (FCS; Sigma-Aldrich, Taufkirchen, Germany), L-glutamine (2 mM), penicillin (100 U/mL), and 100 mg/mL streptomycin (Gibco/Life Technologies). Human iPSCs (BIHi001-B) were cultured in StemFlex Medium (ThermoFisher Scientific, Waltham, MA, USA) in Vitronectin (5 µg/mL, Gibco/Life Technologies) coated wells in a 12-well setting. Passaging of hiPSCs was done by detaching the cells using Accutase (Sigma-Aldrich) supplemented with Y-27632 (10 µM, Tocris Bioscience, Bristol, UK) at a seeding density of 6 × 10^4^/cm^2^. After passaging, the hiPSCs medium was supplemented with Y-27632 (10 µM, Tocris Bioscience) for 24 h. Thereafter, Y-27632 was removed with a full medium change. Medium changes were performed every other day until the well was 70–90% confluent. At this point, the hiPSCs were passaged again.

Cell culture was performed under standard conditions (37 °C, 100% relative humidity, 5% CO_2_). Cell culture material was purchased from Corning (Corning, NY, USA), Greiner Bio One (Frickenhausen, Germany), and Sarstedt (Nümbrecht, Germany).

To generate a stable GFP-overexpressing HEK293T reporter cell line, 50,000 cells were transduced with the γ-retroviral vector RSF91.GFP encoding eGFP [57] at an MOI of 0.1. Subsequently, GFP-positive cells were sorted using a BD FACSAria IIIu sorter (BD Biosciences, San Jose, CA, USA) at the Cytometry and Cell Sorting Core Unit of the UMC Hamburg-Eppendorf and expanded for further experiments.

### 4.2. Molecular Cloning

gRNA-expressing plasmids were generated by using a LeGO-CC vector containing the U6 promoter and gRNA scaffold, and synthesizing the respective DNA sequences (as indicated in Appendix A). If necessary, a “G” was adjoined at the 5′-end of the gRNA/spacer sequence, which is required for polymerase-III-dependent transcription. During synthesis, an ACC triplet and an AAC were added at the 5′ ends of the leading and the complementary strands, respectively, to allow for ligation into the SapI cloning site of the LeGO-CC vector (based on LeGO-iC2, Addgene #27345 [58]).

### 4.3. RVLP Production and Cell Transduction

To produce RVLPs, 293T cells were co-transfected with different combinations of packaging plasmids pcDNA3.MLVgp [59], pMDLg/pRRE [60], and phCMV-VSV-G [61] following our standard protocols for lentiviral and γ-retroviral vectors (Appendix A) [58,62]. Plasmids encoding the gRNA and gag-spCas9 fusion proteins were added, as indicated in Appendix A. The latter, pCMV-MMLVgag-3xNES-Cas9 (Addgene plasmid #181752), pJRH029 (Addgene plasmid # 171060), and BIC-Gag-Cas9 (Addgene plasmid #119942), were kind gifts from David Liu, Jennifer Doudna and Philippe Mangeot, and Théophile Ohlmann and Emiliano Ricci, respectively. Plasmid amounts used for process optimization are indicated in Table 1 in the Section 1.

For production in 100-mm dishes, 5 × 10^6^ 293T cells were plated approximately 16 h before transfection. For downscaled and upscaled productions, the numbers of seeded cells and plasmids were adjusted to the effective growth area. Transfections were performed in a VLP production medium (HEK293T culture medium supplemented with Chloroquine (25 mM)). After 6 h of incubation at 37 °C, the VLP production medium was exchanged. The supernatant was harvested 44–48 h after transfection and filtered (0.45 µm). For concentration, the supernatant was centrifuged overnight at 8000× *g* at 4 °C. RVLP samples were stored at −80 °C.

For transduction, 5 × 10^5^ HEK293T or hiPS cells were plated in 1.0/0.5 mL of appropriate medium in 24-/12-well plates, respectively. At least 4 h after plating, the respective amounts of RVLP-containing supernatant were added. To enhance transduction of 293 T cells, polybrene (8 μg/mL) was added together with the supernatant, and plates were centrifuged at 1000× *g* for 1 h. After 24 h of transduction, cells were washed with PBS (Gibco/Life Technologies), and fresh medium was added.

### 4.4. Flow Cytometry (FC)

RVLP-treated GFP-293T cells were dissociated with 0.5% trypsin-EDTA (ThermoFisher Scientific), and a washing step was performed by centrifugation at 310× *g* for 5 min and resuspension of the pellet in PBS. FC analysis was carried out using a NovoCyte Quanteon Flow Cytometer (Agilent Technologies, Santa Clara, CA, USA). Data were analyzed using BD FACSDiva and FlowJo (both BD Biosciences).

### 4.5. Nanoparticle Tracking Analysis (NTA)

Concentration, size distribution, and mode size of RVLPs were determined by NTA using the Nanosight LM14 instrument (Malvern Panalytical, Kassel, Germany) equipped with a 638-nm laser and a Marlin F-033B IRF camera. Collected RVLP samples were diluted 50- to 1000-fold in PBS prior to NTA analysis. Five 30-s movies were recorded on camera level 16, and then analyzed with detection threshold 6 in NTA 3.2 Build 16. All NTA EV size data are presented as mode values.

### 4.6. ELISA

We established an indirect ELISA based on the assay described by Gutierrez-Guerrero et al., 2021 [44]. Importantly, based on our own preliminary data (Appendix A), we did not use Triton X-100 for the coating process, which is divergent from the described protocol [44]. For each scale of RVLP production (6-well, 100-mm, and 150-mm dishes), dilution of RVLP samples for measurement and the generation of the standard curve were adapted: RVLP samples produced in a 6-well, 100-mm dish, and 150-mm dish were diluted 1:10, 1:50, and 1:100 in PBS, respectively. For each scale of production, three standard curves were generated by adding recombinant Cas9 to final concentrations of 4, 2, 1, 0.5, 0.25, and 0.125 ng/mL (Appendix A). To calculate the amount of Cas9 in a VLP sample, the corresponding standard curve for the given scale of production was used. For coating, we added 100 µL sample per well of the 96-well plate (655081; Greiner Bio-One) followed by incubation at 4 °C for 17 h. After a single washing step with 300 μL washing buffer (PBS/0.05% Tween), the wells were blocked with 200 μL blocking buffer (ROTI-Block, Carl Roth, Karlsruhe, Germany) at 21 °C for 7 h and subsequently incubated with 100 μL diluted (s. below) anti-Cas9-antibody (mouse, Cas9-7A9-3A3 #14697; Cell Signaling Technology, Danvers, MA, USA) at 21 °C for 1h. Four washing steps with 300 μL washing buffer (PBS/0.05% Tween) were performed, and wells were incubated with 100 μL diluted HPR-conjugated anti-mouse-IgG-antibody (W4021; Promega, Fitchburg, WI, USA) at 21 °C for 1 h, again followed by four washing steps with 300 μL washing buffer (PBS/0.05% Tween). For Cas9 quantification, 100 μL of TMB substrate solution (G7431; Promega) were added to the wells for 20 min at 21 °C with subsequent addition of 100 μL stop solution (#7002; Cell Signaling Technology). Finally, optical density was measured at 450 nm using a plate reader (CM Infinite Mono 200, Tecan, Switzerland). Before use, anti-Cas9-antibody and anti-mouse-IgG-antibody were diluted 1:1000 and 1:2500 with dilution buffer (PBS with 3% BSA).

### 4.7. Imaging Flow Cytometry (IFC)

For multi-parametric IFC analysis, RVLPs were produced and purified as described before but with exosome-depleted VLP production medium (Thermofisher Scientific #A2720801). Staining was done by adding 75 µL gRVLP/LVLP supernatant, 3 µL filtered PBS containing 8% exosome-depleted FCS, and 3 µL of each respective antibody followed by 1 h incubation at 21 °C in the dark. For tetraspanin staining, pre-conjugated 1:30 diluted, anti-CD9-PE (#312106, Clone H19a, Biolegend, San Diego, CA, USA) anti-CD63-PacificBlue (#353012, Clone H5C6, Biolegend) and anti-CD81-FITC (#349504, clone 5A6, Biolegend) were added. For Cas9 staining, pre-conjugated anti-Cas9-AlexaFluor488 (#34963, 7A9-3A3, Cell Signaling Technology) was added. For control purposes, VLP were lysed with 50 µL of 10% NP-40 lysis and incubated for 30 min at 21 °C, as described previously [63]. A washing step was done using a 300-kDa filter (OD300C33, Pall Corporation, Port Washington, NY, USA), followed by resuspension in filtered PBS containing 2% exosome-depleted FCS. Measurement was performed on an AMNIS ImageStream^x^ Mk II Flow Cytometer (Luminex Corporation, Austin, TX, USA). Fluorescent signals were collected as follows: PacificBlue was detected in channel 7 (435–505 nm filter), FITC was detected in channel 2 (480–560 nm filter), and Phycoerythrin (PE) was detected in channel 3 (560–595 nm filter). All readings were acquired at 60× magnification and a low flow rate. For data analysis, IDEAS software v6.2 (Luminex Corporation) was used. The gating strategy is shown in Appendix A.

### 4.8. Digital PCR (dPCR)

For subsequent dPCR, 293T and iPS cells were harvested, and genomic DNA was purified with the DNeasy Blood & Tissue Kit (QIAGEN, Hilden, Germany) and quantified with a NanoDrop Spectrophotometer (ThermoFisher Scientific). DPCRs were designed, prepared, and carried out essentially as described in our published protocol [37,56]. In short, PCR mixtures were assembled with 1× ddPCR Supermix for Probes (No dUTP) (#1863024, Bio-Rad, Feldkirchen, Germany), dPCR CCR5 forward primer and dPCR CCR5 reverse primer, CCR5ref and CCR5mut probes, and 40 ng of genomic DNA. The primer and probe sequences are listed in Appendix A. DG8 cartridges (Bio-Rad) were filled with ddPCR reaction mixtures and Droplet Generation Oil (Bio-Rad Laboratories GmbH, Hercules, CA, USA). Droplets were generated with a QX100 Droplet Generator (Bio-Rad) according to the manufacturer’s instructions and then transferred into a 96-well PCR plate for PCR using a T100 Thermal Cycler (Bio-Rad). Cycling conditions were as follows: step 1: 95 °C for 10 min; step 2: 94 °C for 30 s; step 3: 60 °C for 3 min (steps 2 and 3 were repeated 40 times); step 4: 98 °C for 10 min. Droplets from each well were analyzed with the QX100 Droplet reader (Bio-Rad), and the data obtained were analyzed with Quantasoft software version 1.7.4 (Bio-Rad). Negative and positive droplets were discriminated by manual thresholding according to the wt controls included in each individual experiment.

### 4.9. Quantification and Statistical Analysis

If not specified otherwise, datasets shown as bar graphs represent the average of three independent experiments, with error bars indicating standard deviation (SD). Statistical significance was determined with a one-way ANOVA or unpaired Student’s *t*-test with GraphPad Prism (GraphPad Software, version 8.0.1; Boston, MA, USA).

## 5. Conclusions

In conclusion, our data confirm the substantial impact of the type and design of retrovirus-based VLPs on their functioning as GE vectors. In direct comparison, MLV-based egRVLPs as introduced by Banskota et al. [32] showed the best performance of the tested systems. We have also demonstrated that Cas9 content as measured by ELISA and particle numbers as quantified by NTA correlate with actual gene-editing rates conferred by the different RVLPs. We suppose that these two parameters, used in concert, might be very useful to characterize retrovirus-based VLPs.

## Figures and Tables

**Figure 1 ijms-24-11399-f001:**
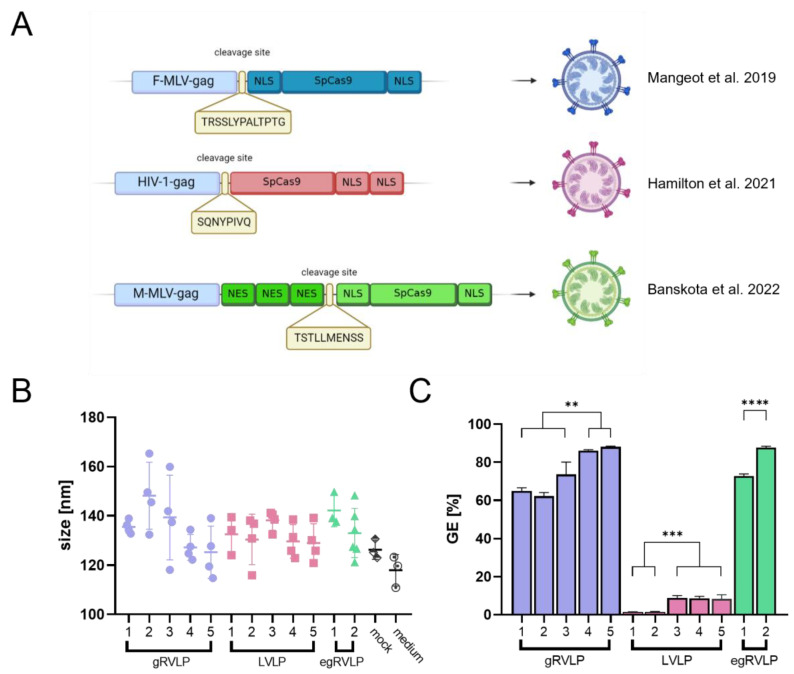
Production optimization and general characterization of three RVLP systems. (**A**) Schematic representation of the molecular design of gRVLP [30], LVLP [31], and egRVLP [32]; created with BioRender.com (**B**) Sizes of the RVLPs obtained for the different GagCas9-to-GagPol ratios (s. Table 1) as measured by the NTA. Data are presented as mean ± SD, n ≥ 3. (**C**) Maximal GE efficiency achieved with VSV-G pseudotyped, GFP-directed RVLPs produced using different GagCas9-to-GagPol ratios (the groups are the same as in (**B**) and Table 1). GE rates were determined after transduction of a dedicated GFP-expressing 293T reporter cell line. For each group, the highest obtained GE rates are shown, representing mean values of at least three independent transductions. Detailed data are presented in Appendix A. NLS = nuclear localization site, NES = nuclear export signal. (** *p* < 0.01, *** *p* < 0.001, one-way ANOVA; **** *p* < 0.0001; unpaired *t*-test).

**Figure 2 ijms-24-11399-f002:**
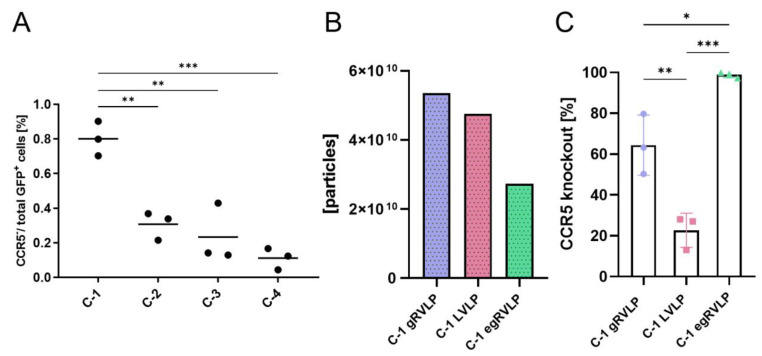
CCR5 targeting with Cas9-VLP systems. (**A**) CCR5 GE efficiencies in 293T cells mediated by integrating all-in-one LeGO-CC vectors programmed with gRNAs C1–C4 (sequences are provided in Appendix A). Indicated is the portion of CCR5-negative cells as measured by GEF-dPCR [37] related to the number of transduced (GFP+) cells. Individual and mean values are shown for three independent experiments. (**B**) Number of particles applied to achieve CCR5-knockout rates presented in (**C**). Particle numbers were measured using NTA. (**C**) CCR5 knockout rates achieved in 293T cells using gRNA C-1 in gRVLP, LVLP, and egRVLP. Data are shown as the percentage of CCR5^ko^ cells in the population. Bars represent mean values ± SD; individual data points are indicated (n = 3). * *p* < 0.05, ** *p* < 0.01, *** *p* < 0.001; one-way ANOVA. Color coding and symbols are as in Figure 1.

**Figure 3 ijms-24-11399-f003:**
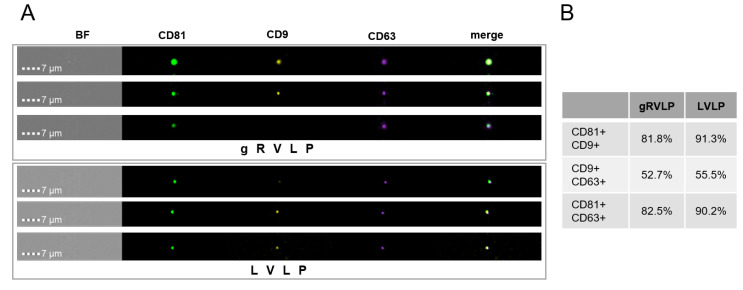
Morphology and tetraspanin marker composition of Cas9-VLPs. (**A**) IFC images of the gRVLPs and LVLPs. Merged pictures show the co-localization of the signals. (**B**) The chart enumerates the relative proportions of VLPs positive for the indicated tetraspanins. BF: bright field.

**Figure 4 ijms-24-11399-f004:**
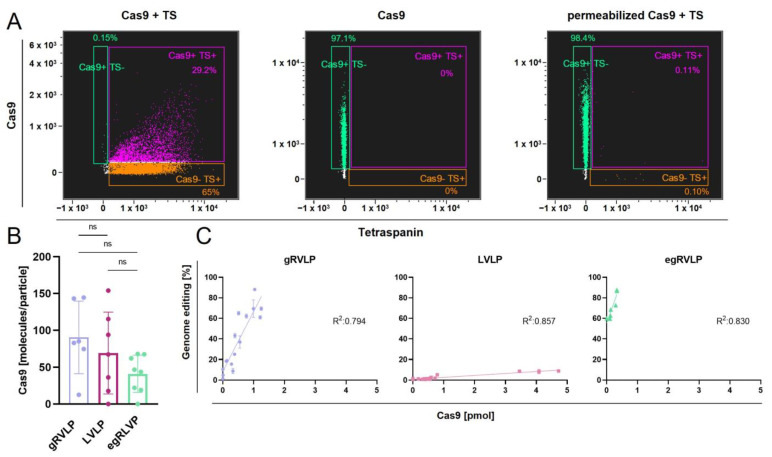
Quantification of the Cas9 content of Cas9-RVLPs. (**A**) IFC analysis of Cas9 presence within Cas9-gRVLPs. On the left dot plot, VLPs were co-stained with a Cas9 (*Y*-axis) and a mixture of tetraspanin (TS, *X*-axis) antibodies. The middle dot plot shows the results of VLPs staining only with a Cas9 antibody. On the right dot plot, VLPs were first lysed and then co-stained with the Cas9 and the TS antibodies. Colored gates demarcate the different single- and double-positive VLP populations. (**B**) Quantification of the Cas9 amount within the RVLPs measured with a dedicated indirect ELISA. Mean numbers of Cas9 particles (bars) ± SD are shown for the three types per VLP for at least six independent experiments. Individual values are indicated as dots. ns = not significant (**C**) Correlation of Cas9 contents measured by indirect ELISA with GE rates achieved with the different VSV-G pseudotyped, GFP-programmed VLPs. The line represents a linear regression. (n ≥ 5). Color coding and symbols in (**B**,**C**) are as in Figure 1.

**Figure 5 ijms-24-11399-f005:**
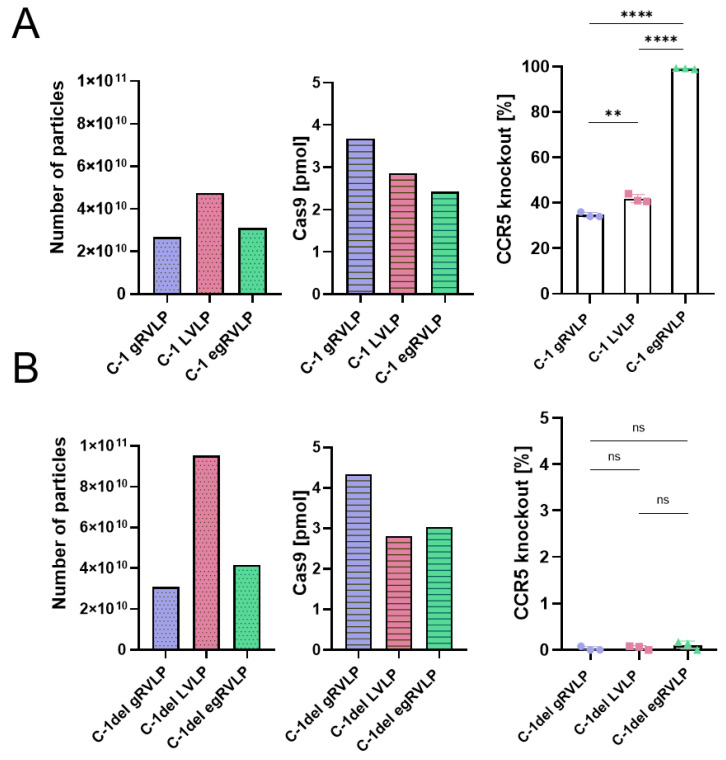
On- vs. off-targeting CCR5 in hiPSC using different retroviral VLP systems. (**A**) CCR5 on-target editing mediated by C-1 programmed gRVLP, LVLP, and egRVLP. The left and middle panels indicate particle numbers and Cas9 contents of the different VLPs measured by NTA and ELISA, respectively. The right panel illustrates CCR5ko rates as measured by GEF-dPCR (mean values ± SD, n = 3, ** *p* < 0.01, **** *p* < 0.0001; one-way ANOVA). (**B**) CCR5 off-target editing mediated by C-1del programmed gRVLP, LVLP, and egRVLP. Again, the left and middle panels show the particle numbers and Cas9 contents of the different VLPs. The right panel depicts off-target CCR5ko rates as measured by GEF-dPCR (mean values ± SD, n = 3, ns = *p* > 0.05, one-way ANOVA). Color coding and symbols are as in Figure 1.

**Table 1 ijms-24-11399-t001:** Plasmids, their quantities and applied GagCas9-to-GagPol ratios used for transfection of 293T producer cells in order to obtain the different types of RVLPs. The amounts were adjusted to the surfaces of single wells of 6-well plates used for RVLP production. Groups 1–5 and 1–2 for the different vectors are the same as in Figure 1. The ratios tested in the original papers are highlighted in bold.

	gRVLP					LVLP					egRVLP	
	1	2	3	4	5	1	2	3	4	5	1	2
gRNA [µg]	1.5	1.5	1.5	1.5	1.5	1.5	1.5	1.5	1.5	1.5	0.7	1.5
GagPol [µg]	0.9	0.9	0.9	0.9	0.9	1.2	0.9	0.6	0.3	1.5	0.5	0.9
GagCas9 [µg]	0.3	0.6	1	2	3	0.3	0.9	1.2	1.5	3	0.2	3
VSV-G [µg]	0.15	0.15	0.15	0.15	0.15	0.15	0.15	0.15	0.15	0.15	0.06	0.15
**Ratio**	1:3	**2:3**	10:9	20:9	10:3	**1:4**	**1:1**	**2:1**	**5:1**	2:1	**2:5**	10:3

## Data Availability

All data associated with this study are present in the paper or in the Appendix A. All raw data are available upon request.

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
