# Peer review of "Deep Characterization and Comparison of Different Retrovirus-like Particles Preloaded with CRISPR/Cas9 RNPs"

_ijms, 2023, doi:10.3390/ijms241411399_

Round 1

Reviewer 1 Report

In manuscript ijms-2486490 Wichmann et al. explore three different approaches to deliver CRISPR/Cas9 RNP via retrovirus-like particles. In this setting, enhanced gamma MLV particles (egRVLP) clearly proved to be the most efficient vectors. The study is certainly of interest for the gene editing field, and is well designed, conducted and written. However, I still have some minor comments which the authors should try to address.

2.2 The authors did not comment on the observation that GE percentage appears to be negatively correlated with egRVLP particle numbers (Fig. S1). What is their explanation for this somewhat unexpected result?

2.4 The authors state that the VLP preparations are of high purity with low contamination by classical EVs based on CD9, CD63 and CD81. How were they able to discern between these two classes? Did they also analyze EVs from mock transfected cells in this respect? Percentages of cells within the IFC gates should be added to Figures 4 and S4.

2.5 Did the authors also perform immunofluorescence to determine the intracellular localization of Cas9 protein in producer as well as recipient cells? According to their hypothesis the superiority of egRVLP is due to enhanced cargo release and nuclear export (or do they rather mean nuclear import in recipient cells?) of Cas9 which could be strengthened by such experimentation.

2.6 Though the authors tried to address off-target editing by usage of a guide carrying a single nucleotide deletion, the outcome of this experiment does not necessarily allow conclusions about off-target indels of the unmutated guide (which would be of potential therapeutic interest). This should be stated in the text.

Furthermore, an outlook on whether this approach may be also suitable for more recent advances (CRISPRi or -a, base and prime editing) should be given in the discussion.

The quality of English is appropriate.

Author Response

Please find a detailled response in the attached document

Reviewer 2 Report

This is a very polished manuscript describing a three-way comparative study of retroviral transduction of Crispr/Cas gene editing components. The work is interesting and well-formulated to assess differences in VLP behaviors relevant to efficient gene editing with minimal off-target effects. The experimental results are clearly presented and adequately discussed in the text. Differences between the three VLPs are convincing.

Author Response

(The authors gave the same response as above.)
